# Dissecting diazirine photo-reaction mechanism for protein residue-specific cross-linking and distance mapping

Yida Jiang [1], Xinghe Zhang[1], Honggang Nie[1,2], Jianxiong Fan[1], Shuangshuang Di[1,2], Hui Fu[1,2], Xiu Zhang[1,2], Lijuan Wang[1,2] & Chun Tang [1,3] ✉

While photo-cross-linking (PXL) with alkyl diazirines can provide stringent distance restraints and offer insights into protein structures, unambiguous identification of cross-linked residues hinders data interpretation to the same level that has been achieved with chemical cross-linking (CXL). We address this challenge by developing an in-line system with systematic modulation of light intensity and irradiation time, which allows for a quantitative evaluation of diazirine photolysis and photo-reaction mechanism. Our results reveal a two-step pathway with mainly sequential generation of diazo and carbene intermediates. Diazo intermediate preferentially targets buried polar residues, many of which are inaccessible with known CXL probes for their limited reactivity. Moreover, we demonstrate that tuning light intensity and duration enhances selectivity towards polar residues by biasing diazo-mediated cross-linking reactions over carbene ones. This mechanistic dissection unlocks the full potential of PXL, paving the way for accurate distance mapping against protein structures and ultimately, unveiling protein dynamic behaviors.

Diazirine is a widely used functional group in photochemistry[1–5], as its three-membered ring readily opens up upon irradiation[6–9]. In chemistry, diazirine has been used to modify small molecules and to build polymers[10,11]. Diazirine's photo-reaction is triggered upon the irradiation at a wavelength of ~365 nm[12], which is routinely employed in biological and chemical experiments[13–21]. As a result, diazirine chemistry has been employed in a multitude of biological applications, most notably, photo-labeling and photo-cross-linking (PXL) of proteins.

Photo-labeling is a process to conjugate a probe to a protein with irradiation, which has been used for receptor identification and drug discovery[22–25]. In PXL, a photo-cross-linker reacts with two protein residues in the vicinity[26–29]. PXL differs from photo-labeling as the probe has a second functional group for protein conjugation. For example, NHS-diazirine (SDA) is a common photo-cross-linker[28], which contains an N-hydroxy-succinimide (NHS) group for chemical reaction with protein primary amine group and a diazirine group for photo-reaction.

The development of chemical cross-linking (CXL) has greatly contributed to the recent progress of structural proteomics. The probe used in CXL contains two chemically reactive functional groups for protein conjugation, e.g., the two NHS groups in bis-sulfo-succinimidyl suberate (BS$_3$)[30]. Subsequently, the cross-linked peptides are identified by using high-resolution mass spectrometry[31–35]. For its easy implementation and standardized analysis workflow[36], CXL has been increasingly used to assess the structure and dynamics of proteins and protein complexes[37–43]. However, CXL reactions typically occur in minutes, much slower than photo-reactions[38]. Moreover, the inter-residue distances from CXL are usually longer than those from PXL[1,28,38,44,45], thus providing only weak structural constraints. Nevertheless, PXL reactions can be highly heterogeneous, yielding a mixture of products with protein residues[23,46–50]. The lack of a detailed and quantitative understanding of the photo-reaction mechanism and products has prevented the wide use of PXL for protein structure analysis.

[1]Beijing National Laboratory for Molecular Sciences, College of Chemistry and Molecular Engineering, Peking University, Beijing, China. [2]Analytical Instrumentation Center, College of Chemistry and Molecular Engineering, Peking University, Beijing, China. [3]Center for Quantitative Biology, PKU-Tsinghua Center for Life Sciences, Academy for Advanced Interdisciplinary Studies, Peking University, Beijing, China. ✉e-mail: Tang_Chun@pku.edu.cn

Diazirine photo-reaction is the key to the success of photo-labeling and PXL applications[18]. Fluorine-substituted aryl diazirines upon irradiation have been shown to yield carbene intermediates that can be inserted into C-H or N-H bonds of adjacent residues[19,51]. Alkyl diazirines, in comparison, are less bulky and less hydrophobic than aryl ones. Sinz and coworkers have shown that in addition to the carbene intermediate, irradiation of alkyl diazirine can give rise to a diazo intermediate that can react with acidic residues[48]. More recently, Woo and coworkers have shown that diazirine-based compounds are preferentially labeled towards protein acidic patches in a pH-dependent manner[47]. As both diazo and carbene intermediates can be derived from diazirine upon irradiation[10,52–55], how the two intermediates are generated, their relative yields and their preferences toward protein residues are yet to be established.

Commercially available lightbox typically features one or two broad-wavelength mercury bulbs with limited irradiation power. Moreover, $[h\nu]$ cannot be varied, and the relationship between photo-reaction and optical power density cannot be evaluated. These limitations hinder a deeper understanding of the photo-reaction mechanism of alkyl diazirine.

In this work, we build an instrument equipped with an array of LEDs and in-line mass spectrometry monitoring. Using this setup, we obtain detailed kinetic parameters and demonstrate distinct preferences of diazo and carbene towards protein side chains, thus allowing protein structure evaluations with residue-specific PXLs.

## Results

### Uncovering diazirine photolysis mechanism with a power-modulated photo-reaction system

When absorbing a photon, the diazirine group undergoes photolysis through one of the four mechanisms, as shown in Fig. 1. Alkyl diazirine **A** transforms to diazo intermediate **B** or directly to carbene intermediate **C**, with the kinetic rate constants of $k_1[h\nu]$ and $k_3[h\nu]$, respectively. **B** absorbs a second photon and transforms to **C**, with the kinetic rate constant of $k_2[h\nu]$. Thus, model **I** is a simplified version of model **II**, differing in the lack of a direct **A**-to-**C** process. Model **III** assumes that the **B**-to-**C** process occurs spontaneously without irradiation, while model **IV** involves no **B**-to-**C** process.

At a given $[h\nu]$, theoretical curves for these four models can be plotted, in which the concentrations of **A**, **B**, and **D** exhibit time-dependent changes (Fig. 1e–h). Model **I** differs from model **II** with a lag in the buildup of **D**, while model **IV** predicts a single-exponential curve of **D**. Model **III** predicts a time-dependent change of **D** that is sensitive to optical power density $[h\nu]$, and, therefore, would appear differently at different $[h\nu]$ (see Supplementary Note).

To identify which model best describes the photolysis mechanism of alkyl diazirine, we built a real-time photo-reaction system coupled with in-line mass spectrometry (MS) detection (Supplementary Fig. 1a). In this MS setup, the sample is injected with a constant flow into a PFA tube arranged zigzaggedly and reacts continuously under irradiation. An array of light-emitting diode (LED) at the 365 nm wavelength was arranged to produce a uniform irradiation field covering the PFA tube (Supplementary Fig. 1b). The reaction mixture was subsequently injected into a coupled triple-quadruple MS for real-time multiple reaction monitoring (MRM) analysis, allowing for highly specific and sensitive label-free quantification.

A pulse-width modulation scheme was used to manipulate the photo-reaction time $t$, as the sample flows through the PFA tube of a fixed length at a constant flow rate (Supplementary Fig. 1c). As the photo-reaction occurs in real-time with a constant flow scheme, no internal standard is needed[56–58]. On the other hand, the optical power density $[h\nu]$ can be adjusted, providing a second dimension for the establishment of the kinetic mechanism (Supplementary Fig. 1d). Together, a large number of experimental observables can be obtained.

### Diazo not carbene is the main intermediate upon alkyl diazrine photolysis

To dissect the photolysis mechanism of alkyl diazirine. We used sulfo-SDA, for its extra sulfate group and excellent MRM signal in the negative ion mode. We monitored the photo-reaction in real-time with MRM-MS and found that the time-dependent increase of the MS signal for **D** cannot be fit to a single-exponential function (Fig. 2a), which allowed us to exclude model **IV**. On the other hand, increasing optical power density $[h\nu]$ from 101 mW/cm² to 242 mw/cm² caused little change to the profile of **D** (Supplementary Fig. 2), which led to the exclusion of model **III**.

In either model **I** and model **II**, the diazo intermediate **B**, isomeric to **A**, is quickly converted to the carbene intermediate **C** with the loss of a nitrogen molecule. The production and subsequent disappearance of diazo intermediate **B** can be monitored with in-line NMR spectroscopy based on the characteristic peak of the methyl group (Supplementary Fig. 3). Moreover, the diazo intermediate can be captured with methacrylate to generate a pyrazole product that can be confirmed with MS (Supplementary Fig. 4). To further distinguish between model **I** and model **II**, we assessed the time-dependent change of [**A**], which should follow a single-exponential decay (Fig. 1e, f), while the exponent $(k_1 + k_3)[h\nu]$ could be determined by varying optical power density $[h\nu]$ (Fig. 2b). We also determined the second exponent $k_2[h\nu]$ with linear regression over different optical power density, and found it smaller than $(k_1 + k_3)[h\nu]$ (Fig. 2b).

We determined the values of $k_1[h\nu]$ and $k_3[h\nu]$ by varying $[h\nu]$ and obtained the ratio of $k_1/(k_1 + k_3)$ (Fig. 2c). The $k_1/(k_1 + k_3)$ ratio decreases slightly from about 0.92 to 0.85 at increasing optical power density $[h\nu]$, meaning that an **A**-to-**B** process is dominant for the consumption of alkyl diazirine. As such, model **II** best describes the photolysis mechanism of sulfo-SDA (Fig. 1).

### Alkyl diazirine can selectively react with polar residue

Reactions were performed between SDA and AXA tripeptide (also denoted as **HY**, for the elementary reactions discussed below), with X representing any amino acid (Fig. 3a), at different optical power density $[h\nu]$ and irradiation time $t$. The tripeptide is N-terminally acetylated and C-terminally methylated (Supplementary Fig. 5), and therefore, the residue X largely mimics that in a protein[59,60]. The production of **SDA-HY** is mediated by either diazo or carbene intermediate, which comprises five elementary processes defined as $[\mathbf{SDA-HY}]_{\{[h\nu],t\}} = [\mathbf{SDA-HY}]^{\mathrm{B}}_{\{[h\nu],t\}} + [\mathbf{SDA-HY}]^{\mathrm{C}}_{\{[h\nu],t\}} = b_1 I_{b_1} + b_2 I_{b_2} + b_3 I_{b_3} + b_4 I_{b_4} + c I_c$. The reaction involving the diazo intermediate **B** proceeds through one of four mechanisms, in which $I_{b_1}$ and $I_{b_2}$ represent the direct reaction between the protonated form of the tripeptide **HY** and **B** and between the deprotonated form of **Y**⁻ and **B**, respectively, whereas $I_{b_3}$ and $I_{b_4}$ represent the corresponding proton-catalyzed processes, respectively. The carbene intermediate **C** reacts with **HY** via carbene insertion, and $I_c$ is the corresponding function for the production of **SDA-HY**, related to optical power density $[h\nu]$ and irradiation time $t$.

With a fixed irradiation time $t$, all four sub-reactions of **B** maximize at a particular optical power density. This is because a larger $[h\nu]$ causes more conversion of **B** to **C**, thus decreasing the concentration of **B**. The maximum peaks are more pronounced for proton-catalyzed $I_{b_3}$ and $I_{b_4}$ processes than non-proton catalyzed $I_{b_1}$ and $I_{b_2}$ processes (Fig. 3b). On the other hand, carbene-mediated reaction $I_c$ increases monotonically with $[h\nu]$. Thus, owing to the different dependence over optical power density, the relative contributions of these five elementary processes, $b_1$, $b_2$, $b_3$, $b_4$, and $c$ values, for the production of **SDA-HY** can be evaluated experimentally.

The experimental data and fitted curves for ASA and AIA tripeptides are shown in Fig. 3c, and all other residues in Supplementary Fig. 6; the fitted values of $b_1$, $b_2$, $b_3$, $b_4$, and $c$ are provided in Supplementary Table 1. A positive value of $(b_1 + b_2 + b_3 + b_4 - c)$ indicates a

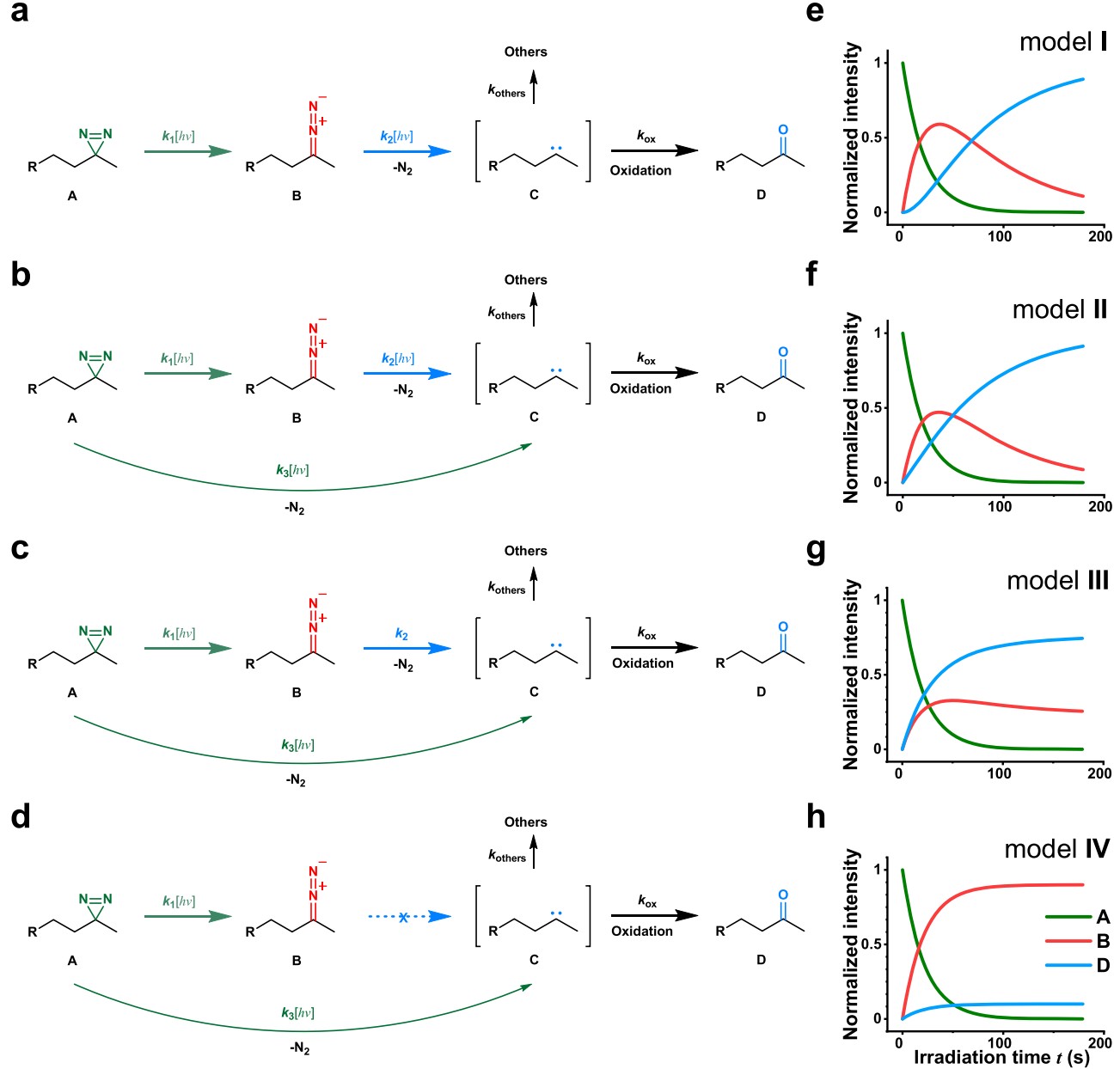

**Fig. 1 | Possible mechanism for the photolysis of alkyl diazirine.** (a–d) Four possible models on the basis of previous findings, inter-conversion pathways, and kinetic mechanisms. Diazo **B** and carbene **C** are the two intermediates that can be generated from the alkyl diazerene **A** upon irradiation, which eventually yields **D**. (e–h) Time-dependent concentration changes of **A**, **B**, and **D** are plotted for these four models. Here, $[A]_t = [A]_0 \exp\left(-(k_1 + k_3)[h\upsilon]t\right)$,

$[B]_t = \frac{k_1[A]_0}{k_1 - k_{2'} + k_3}\left(\exp(-k_{2'}[h\upsilon]t) - \exp(-(k_1+k_3)[h\upsilon]t)\right)$, $[D]_t = \frac{k_{ox}}{k_{ox} + k_{others}}[A]_0$ $\left(1 - \frac{k_1}{k_1 - k_{2'} + k_3}\exp(-k_{2'}[h\upsilon]t) + \frac{k_{2'} - k_3}{k_1 - k_{2'} + k_3}\exp(-(k_1+k_3)[h\upsilon]t)\right)$. In model **I**, $k_3 = 0; k_{2'} = k_2$ except for model **III** where $k_{2'} = k_2/[h\upsilon]$; in model **IV**, $k_2 = 0$. Detailed deduction of the equations can be found in Supplementary Note.

predominantly diazo-mediated production of **SDA-HY**, while a negative value, a carbene mechanism (Fig. 3d). Aliphatic and non-polar residues such as Gly, Ala, Val, Leu, Ile, and Met exhibit $(b_1 + b_2 + b_3 + b_4 - c)$ values close to −1, whereas polar residues such as Ser and Thr residues, close to 1. Note that the hydrophilic residues such as Gln and Asn also exhibit negative values due to the lack of nucleophiles. Interestingly, though the carbene intermediate can react with the Tyr side-chain, the diazo intermediate reacts more preferably either with Tyr-O- or, to a smaller extent, with Tyr-OH but catalyzed by proton. We could identify the carboxylate-modified product of AEA tripeptide with one-dimensional proton NMR for the formation of a new ester bond (Supplementary Fig. 7), while it is not the case for

photo-adduct with AIA tripeptide for the highly heterogeneous carbene insertion (Supplementary Fig. 8).

To what extent the diazo or carbene intermediate is involved in the reaction also determines the overall yield of the photo-adduct. The yield of diazo-mediated **SDA-HY** maximizes at a particular optical power density $[h\upsilon]$, whereas the yield of carbene-mediated adduct increases monotonically with $[h\upsilon]$. Moreover, a long irradiation time $t$, which has been a common practice in batch PXL experiments[26,61,62], would lead to excessive generation of carbene intermediate (Fig. 3b). As such, to improve the selectivity for polar residues, a relatively large optical power density $[h\upsilon]$ and a relatively short irradiation time $t$ should be used (Fig. 4 and Supplementary Fig. 9).

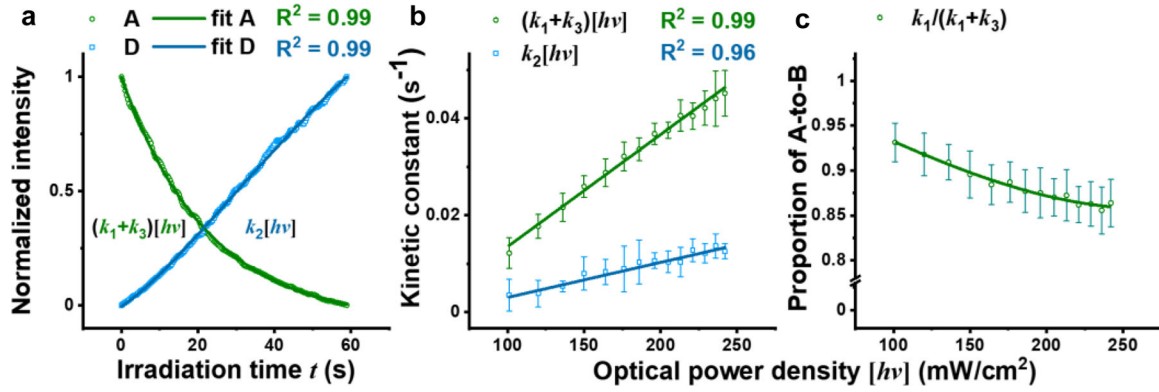

**Fig. 2 | Identification of the kinetic model that best describes diazirine photolysis. a** Fitting the consumption of **A** and the accumulation of **D** over irradiation time $t$, at optical power density $[h\nu]$ of 242 mW/cm², yielding $(k_1+k_3)[h\nu] = (0.0442 \pm 0.0003)$ cm²/(mW·s) and $k_2[h\nu] = (0.0099 \pm 0.0008)$ cm²/(mW·s). **b** Linear regression of $(k_1+k_3)[h\nu]$ and $k_2[h\nu]$ over $[h\nu]$, yielding $(k_1+k_3) = (2.32 \pm 0.06) \times 10^{-4}$ cm²/(mW·s) and $k_2 = (0.73 \pm 0.04) \times 10^{-4}$ cm²/(mW·s). **c** The direct conversion route to carbene intermediate is less opted, as the fitted value of $k_1/(k_1+k_3)$ ranges between 0.85 and 0.92. Each averaged value (open circle) was obtained from triplicated experiments on our real-time photoreaction system coupled with in-line mass spectrometry (MS) detection, with the error bar representing 1 S.D.; the fitted values are also plotted as lines.

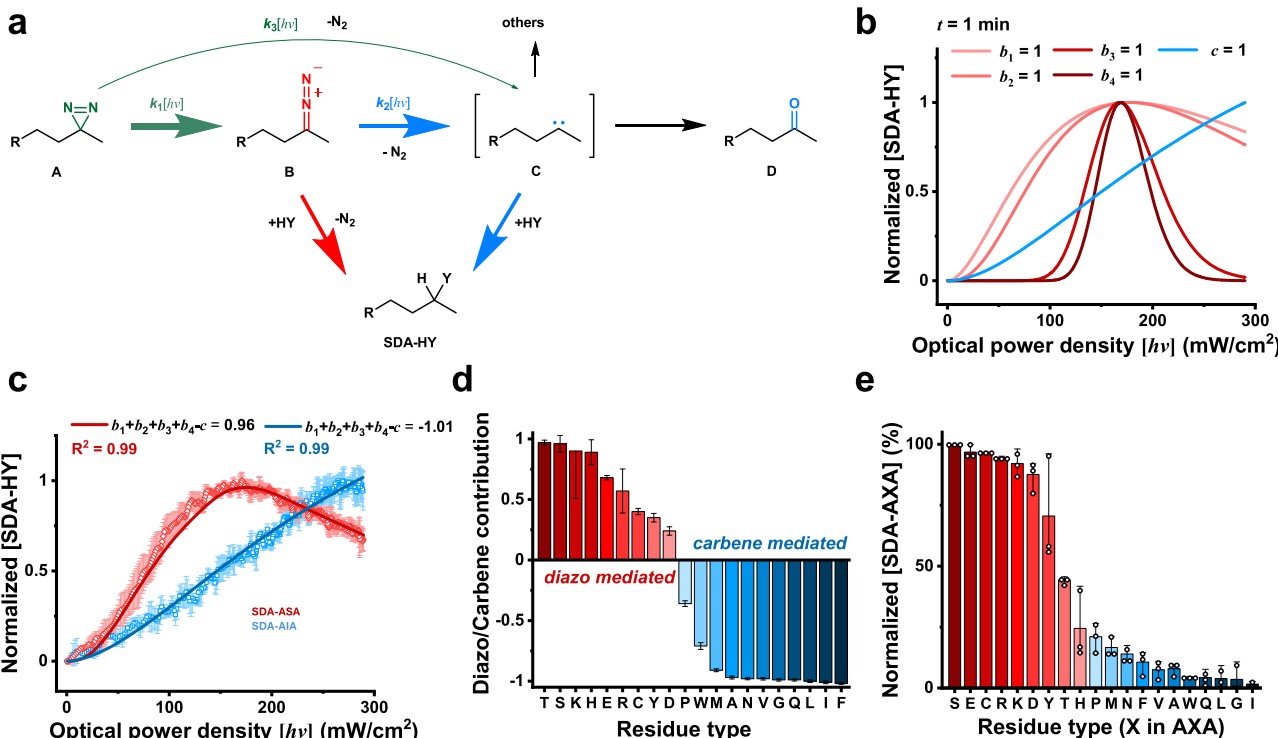

**Fig. 3 | Dissecting the reaction mechanism between alkyl diazirine and protein residues. a** The reaction between alkyl diazirine (SDA) and a protein residue in AXA tripeptide (HY, inset). **b** The generation of **SDA-HY** can be attributed to five elementary processes, $b_1$, $b_2$, $b_3$, $b_4$, and $c$. The theoretical hypothetical curves of these processes are plotted with the irradiation time $t$ of 1 min. **c** Fitting the experimental data of **SDA-HY** production over optical power density $[h\nu]$ allows the determination of the relative contribution of the elementary processes. Shown here are the fitting results for ASA (red) and AIA (blue) tripeptides, with Pearson's correlation $R^2$ of 0.99 for both. Error bar represents 1 S.D. from 3 separate measurements with the averaged values shown as open circles. **d** The relative involvement of diazo **B** or carbene **C** intermediate for the general SDA-HY can be assessed based on the value of $(b_1 + b_2 + b_3 + b_4 - c)$, which can be positive or negative, respectively. Error bar represents 1 S.D. from curve fitting (Supplementary Fig. 6). **e** The relative yield of **SDA-HY** can be evaluated by the decrease of MS signal of AXA tripeptide. Polar residues that are preferentially targeted by diazo intermediates are colored red, and non-polar residues that are preferentially targeted by carbene intermediates are colored blue. Error bar represents 1 S.D. from 3 separate measurements (open circles).

We then assessed the absolute yield of **SDA-HY** based on the relative decrease of peptide MS signal. With the irradiation optimized for polar residues, most polar residues have a yield close to 100% within 2 min of irradiation. In contrast, the conversion rates for non-polar residues are much lower, with the yield of Ile and Val ~50-fold lower (Fig. 3e). It should be noted that though Ala-appended AXA tripeptides are intended to mimic a protein, only enhanced cross-linking above the yield observed for AAA tripeptide can be considered for authentic contribution from the X residue. Interestingly, the yield of the ATA adduct is much lower than other polar residues, which is likely due to steric hindrance from the vicinal methyl group.

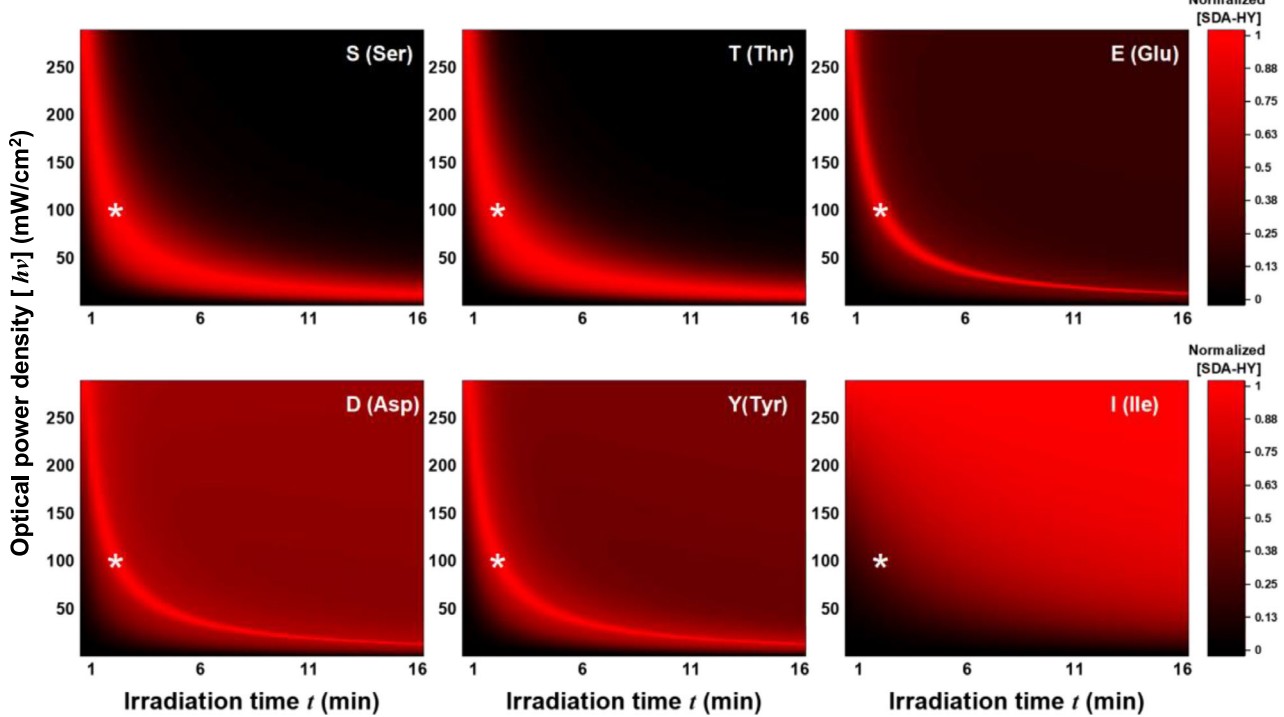

**Fig. 4 | The yield of SDA-HY as a function of irradiation time t and optical power density [*hv*].** For polar residues, maximum yield can be achieved with an optimal combination of *t* and [*hv*]; the asterisks indicates a preferred value for protein PXL experiments (100 mW/cm², 2 min).

We further assessed the conversion rates of tripeptides in water-DMSO mixed solvent. Water readily reacts with either diazo or carbene intermediates, leading to a large drop in the yield of SDA-AXA. Yet, the photo-adduct with polar residues generally gives a higher yield than that with non-polar residues (Supplementary Fig. 10). Note that the yield is almost 0 for ASA adduct at 99% water content, as the Ser hydroxyl group is out-competed by water. Thus, relatively buried residues in a protein should be better placed for diazirine reactions.

### Residue-specific PXL allows for accurate distance mapping of protein structure

We performed PXL experiments using SDA for nine model proteins. The reaction was carried out in two steps: first, the cross-linking reagent reacts with protein Lys residues in the amine-free buffer, and second, irradiation of 365-nm light at a set power initiates photo-reaction between the alkyl diazirine group and an adjacent protein residue. Note that chemical cross-linking not only enhances the local effective concentration for subsequent photo-cross-linking but also partially shields the photolysis intermediate from the solvent.

We assigned the cross-linked peptide spectra using pLink2[63], a search engine that has been primarily used for the analysis of CXL spectra. The abundance of cross-linked Asp, Glu, and Tyr residues far exceeds their natural abundance, especially at a low optical power density [*hv*] (Fig. 5a). The number of peptide-spectrum match (PSM) follows a similar trend, as the percentage of PSM for polar residues generally exceed the corresponding occurrence in the proteins but decreases at a higher optical power density (Supplementary Fig. 11). That the specific PXLs for polar residues are mediated by diazo intermediate was further confirmed with the identification of α ion fragments for the esterified peptide, which could be cleaved from the Lys side-chain of the cross-linked peptides (Fig. 5b and Supplementary Fig. 12).

The differential utilization of the diazirine photolysis intermediate is also evidenced by the analysis of the loop links. The loop-links involve fewer polar residues, which percentage is generally lower than

the corresponding natural abundance (Fig. 5c and Supplementary Fig. 11). This is because many non-polar residues are simply nearby and ready for loop-link reactions, even though carbene intermediate is short-lived and susceptible to water quenching. Interestingly, the loop links involving Tyr are extremely rare, which not only confirms preferential utilization of the diazo intermediate but may also be due to the unfavorable geometry of Tyr bulky side-chain.

We then calculated the Cα-Cα distances of the cross-linked residues based on the protein structures (Fig. 5d). The Cα-Cα distances involving polar residues that readily react with the diazo intermediate are generally within the range permitted by the cross-linker (Supplementary Fig. 13). In contrast, a large proportion of the cross-links involving non-polar residues afford the calculated Cα-Cα distances exceeding the corresponding maximum distance (Supplementary Figs. 14 and 15). Gly, Ala, and Ile residues represent some extreme cases, with the calculated Cα-Cα distances over length by 10 Å or more (Fig. 5d). As the hydrophobic residues are poorly reactive with alkyl diazirine through the carbene mechanism, especially upon water exposure (Fig. 3e and Supplementary Fig. 9), one explanation is that these residues are incorrectly assigned using the search engine designed for CXL. Indeed, the precursor mass errors of the assigned peptides are somewhat larger for cross-links involving non-polar residues than polar residues (Supplementary Fig. 16).

The calculated distances for PXLs involving Glu, Tyr, and Asp are mostly consistent with the protein structures. In contrast, a large proportion of the PXLs involving Thr was found to be over-length (Supplementary Fig. 14 and 15). Moreover, the cross-linked Thr residues are found in highly solvent-exposed regions as compared to the average solvent exposure of Thr in the test proteins (Supplementary Fig. 17). The water-quenching experiment indicated that solver-exposed hydroxyl group is unlikely to out-compete water for the diazo intermediate (Supplementary Fig. 10). Even in a water-free solution, SDA photo-reaction with Thr has a much lower yield than with other polar residues (Fig. 3e). Thus, to minimize false positives, PXLs involving Thr residues are better not used as structural restraints.

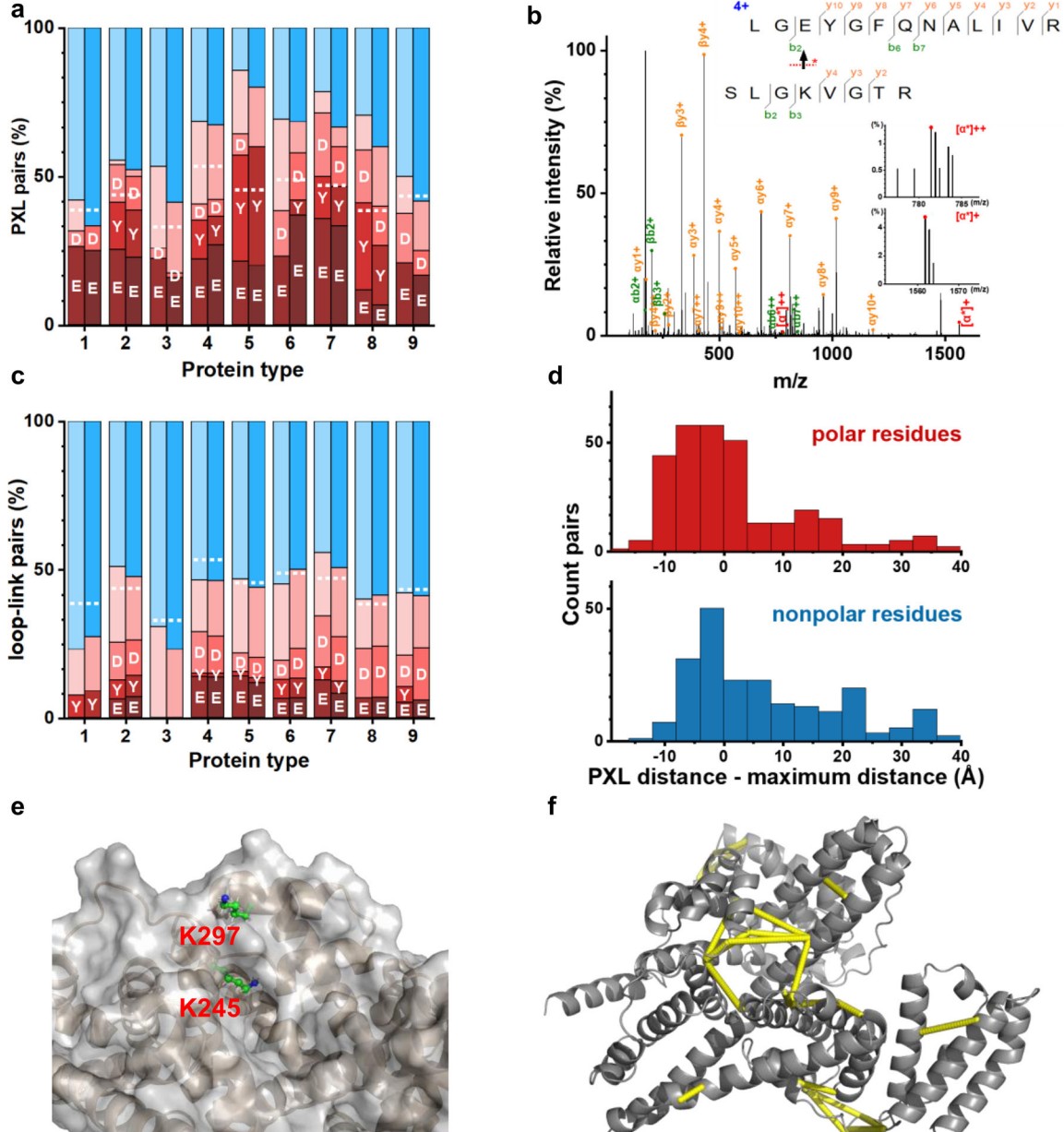

**Fig. 5 | Alkyl diazirine preferentially cross-links polar residues in the 9 test protein. a** Relative abundance of cross-linking pairs for polar residues (red shaded) and non-polar residues (blue shaded), by SDA (first chemically cross-linked to Lys) upon irradiation at an optical power density $[hv]$ of 35 mW/cm² (low-power, left columns, light-colored) for 10 min or 282 mW/cm² (high-power, right columns, dark colored) for 2 min. The dashed line indicates the natural abundance of polar residues in each protein, with D (Asp), E (Glu), and Y (Tyr) labeled. The details of the test proteins, 1–9, are given in "Methods". **b** Fragmentation of the cross-link between Lys and Glu residues (segment of Glu denoted as α) corroborates the assignment. **c** Relative abundance of the loop-links, for non-polar and polar, resides in the test proteins upon low- or high-power irradiation. **d** Normalized Cα-Cα distance distribution (in 4-Å bins) between the cross-linked Lys residue and polar/non-polar residues. Normalization is performed by subtracting the maximally allowed distance (Supplementary Fig. 13) from the calculated Cα-Cα distance. **e** A carved-in structure figure of BSA, illustrating the solvent exposure of residues K245 and K297. **f** Distance mapping from the PXL data involving polar residues onto the BSA structure.

Indeed, relatively buried residues are more likely cross-linked, affording high-quality structural restraints. On one hand, Lys residues with a relatively small solvent exposure are more shielded from the solvent, and therefore, once chemical conjugated with the cross-linker, the diazo intermediate is more likely to cross-link to adjacent polar residues without being quenched. For example, residues K245 and K297 in the BSA are both localized in a helix, with K245 more buried (Fig. 5e). As a result, over twice as many PXL residue pairs and matched spectra are found for K245 than for K297. On the other hand, Tyr residues are often buried and largely shielded from water

(Supplementary Fig. 17), which accounts for the high abundance of Tyr cross-links.

Together, mechanistic dissection of alkyl diazirine photo-reactions significantly expands the repertoire of cross-links, which now includes polar residues of Glu, Tyr, Ser, Asp, as well as Arg and His, though with smaller occurrence. These unique high-quality specific cross-links allow for distance mapping against the protein structure (Fig. 5f and Supplementary Fig. 18), and also allow for the identification of truly over-length cross-links arising from protein dynamics. In fact, a close inspection revealed that the over-length PXLs involving polar

residues could be indicative of transient domain closure of the test proteins[64].

## Discussion

In this work, we have built a power-modulated photochemical reaction system with the optical power density that can be adjusted from 0.1 mW/cm² to nearly 300 mW/cm². For the 365 nm light passing through a sample of 1 cm thickness, about 30 μM/s photons are expected at an optical power density of 10 mW/cm². Considering possible solvent absorption, the density of the photons should be higher than the concentration of the photo-reactive group. As such, our setup enabled us to perform in-line real-time monitoring of photo-reaction, which cannot be done on a commercial one. Using our setup, we have dissected the detailed mechanism of the photolysis of alkyl diazirine and subsequent reactions with protein residues, and obtained residue-specific PXLs that can be mapped to protein structures.

We have shown that upon the irradiation of alkyl diazirine, diazo, and carbene intermediates are generated largely in sequential order. The diazo intermediate preferentially and specifically reacts with protein polar residues. In comparison, the carbene intermediate reacts with non-polar residues, yielding heterogeneous products. The selectivity toward polar residues can be further enhanced with an optimal combination of optical power density $[h\nu]$ and irradiation time $t$, whereas prolonged irradiation would result in an increased formation of carbene intermediate and a decreased selectivity for polar residues.

Through the dissection of the photo-reaction mechanism of alkyl diazirine, we have demonstrated that PXLs for polar residues can be quantitatively analyzed in a similar fashion to what has been established for the analysis of CXLs. Significantly, the cross-linked residues now include Glu, Asp, and Tyr with high confidence. Moreover, the cross-linked residues are relatively buried and thus are likely localized in protein-ordered regions. As CXLs preferentially involve residues in protein-disordered regions[65], the PXLs can provide more stringent constraints of protein structure. On the other hand, metabolically incorporated photo-Leu residues are often deeply buried in the protein hydrophobic core and, therefore, may only react with adjacent hydrophobic residues through the carbene mechanism[1]. This would result in the production of heterogeneous products, as we have shown, making a quantitative interpretation of the PXLs difficult.

The correct assignment of the buried polar residues allows for distance mapping and evaluation of protein structures individually or on a proteomics scale. Residue-specific assignment of PXLs at the protein interface can also be obtained to construct protein complex structures, complementing AI-based structural modeling[43,66,67]. Moreover, a plethora of accurate distance restraints would cross-validate one another, thus enabling the identification of genuinely over-length PXLs incompatible with the known protein structures, which can be a manifestation of protein dynamics[68,69]. Lastly, as photo-reaction can be modulated with temporal precision thanks to the fast reaction kinetics, we envision PXL be used for capturing time-resolved protein intermediary conformations in a reaction trajectory.

## Methods

### Reagents and materials

NHS-diazirine (SDA) was purchased from Bidepharm (Shanghai, China, catalog number 1239017-80-1), and sulfo-NsHS-diazirine (sulfo-SDA) from Thermo Fisher Scientific (Shanghai, China, catalog number 26173). AXA tripeptides (X stands for any amino acid, with N-terminus acetylated and C-terminus methylated) were ordered from TGpeptide Biotechnology (Nanjing, China) for X = Ala, Asp, Asn, Cys and Lys, and Genscript Biotech (Nanjing, China) for the others. The model proteins used include (1) non-structural protein 5, also known as the main protease, from SARS-CoV2 (Uniprot P0DTD1, PDB code 6XB0)[70], (2) Glutathione S-transferase class-mu 26 kDa isozyme (Uniprot ID

P08515, PDB code 1B8X), (3) p27 capsid from Rous Sarcoma Virus (RSV, Unitprot ID P03322, PDB code 7NOO)[71], (4) bovine serum albumin (UniProt ID P02769, PDB code 4F5S, purchased from Sigma Aldrich with catalog number 146897-68-9), (5) lactoferrin (UniProt ID Q6LBN7, AlphaFoldDB AF-Q6LBN7-F1, purchased from Fujifilm Wako Pure Chemical with catalog number 146897-68-9, Tokyo, Japan), (6) monomeric ultra-stable GFP (Uniprot ID P42212, PDB code 5JZK)[72], (7) conalbumin from chicken egg (UniProt ID P02789, PDB code 2D3I, purchased from Sigma Aldrich with catalog number 1391-06-6)[73], (8) proteasomal ubiquitin receptor ADRM1/Rpn13 (UniProt ID Q16186, PDB code 2KR0)[74], (9) adenylate kinase from *E. coli* (UniProt ID P69441, PDB code 1AKE)[64]. The non-commercially available proteins were purified with established protocols in refs. [64,70–72,74,75]., for proteins (1), (2), (3), (6), (8), and (9), respectively.

### Real-time automated photo-reaction with in-line monitoring

The sample was pumped using a Shimadzu Nexera XR LC-30AD Pump in 0.200 mL/min. The solution was injected through the opaque PEEK tube into a transparent PFA tube (1/16 -inch O.D., 0.5 mm I.D.; for a total length of 1 meter, 0.196 mL in volume) for photoreaction. Programmable LED and adjustable constant current supply were customized by Lightwells (Shenzhen, China). Adjustable constant current supply and pulse-width modulated (PWM) signal were controlled with Raspberry Pi 4B (https://www.raspberrypi.com). Upon photo-reaction in the PFA tube, the solution was continuously injected into Shimadzu 8050 MS for real-time multiple-reaction monitoring (MRM) analysis (experimental parameters are provided in Supplementary Tables 2 and 3), following the design illustrated in Supplementary Fig. 1. LabSolutions (Version 5.91) from Shimadzu Corporation was used for data analysis.

Sulfo-SDA was dissolved in 100 mM DMSO, and diluted with acetonitrile to a final concentration of 0.25 μM. The flow rate was set at 0.200 mL/min through the PFA tube, which takes 58.9 seconds (~1 min). The irradiation time $t$ was linearly manipulated by PWM under a constant optical power density $[h\nu]$. The reactant $A$ and product $D$ (Fig. 1) were monitored in the MRM mode, with the total ion chromatogram normalized before fitting. Time-dependent changes of $A$ and $D$ were fitted with linear regression to obtain $(k_1 + k_3)[h\nu]$ and $k_2[h\nu]$; the photo-reactions and measurements were repeated at least three times.

The NMR data was collected on a Bruker Avance III 500 MHz Spectrometer ($B_1 = 500.13$ MHz), equipped with a 5.0 mm Probe head (BBO 500S1 BBF-H-D-05 Z SP). NMR signal was collected in-line with the photo-reaction system; 72 transients were collected, with 57344 points in the time domain, 12 ppm in spectral width, 2.5 ppm for the transmitter frequency offset, 1.0 s in relaxation delay, 4.78 s in total acquisition time, and 64 in receiver gain.

The short-lived diazo intermediate $B$ was captured with methyl methacrylate. SDA and methyl acrylate were dissolved in deuterated DMSO to a final concentration of 10 mM. The solution was irradiated under 2.8 mW/cm² of 365 nm light for 30 min. The captured product, methyl 5-(3-((2,5-dioxopyrrolidin-1-yl)oxy)−3-oxopropyl)−5-methyl-4,5-dihydro-1H-pyrazole-3-carboxylate, was confirmed with the use of Fourier Transform Ion Cyclotron Resonance Mass Spectrometer (Solarix XR, Bruker), affording m/z of 312.118 ([M + H]⁺), 334.101 ([M + Na]⁺), and 350.075 ([M + K]⁺).

SDA and AXA tripeptides [HY] were separately dissolved in 100 mM DMSO and diluted with acetonitrile to a final concentration of 5 μM and 1 μM, respectively, which were then mixed. Note that all the peptides can be readily dissolved. The solution was irradiated with optical power density of 100 mW/cm² for 2 min. The concentrations of [HY] were monitored in Shimadzu 8050 MS in MRM analysis in real-time as described. The yield was determined through the conversion rate or the decrease of MRM signal of HY. The measurement was repeated three times for each peptide. To assess the water quenching

effect, water was mixed with DMSO to the desired water content (v/v) for photo-reaction.

### Mathematic analysis for the kinetic parameters of diazirine photolysis

The normalized intensity of $A$ (sulfo-SDA) over irradiation time $t$ was fitted with $I_A = [A]_0 \exp(-(k_1 + k_3)[h\nu]t) + I_0$, in which $[A]_0$ is the initial concentration and $I_0$ is the offset. The detailed deduction for the production of $D$ is provided in the Supplementary Note. At an irradiation time $t \geq 20$ s, with the assumption of $k_2 < k_1 + k_3$, the real-time concentration of $D$ can be represented as $I_D = [A]_0(1 - D_0 \exp(-k_2[h\nu]t)) + I_0$.

Upon obtaining $(k_1 + k_3)[h\nu]$ and $k_2[h\nu]$, $(k_1 + k_3)$ and $k_2$ was fitted with linear regression by varying optical power density $[h\nu]$, as shown below

$$I_D = [A]_0 \left(1 - \frac{k_1}{k_1 - k_2 + k_3} \exp(-k_2[h\nu]t) \right.$$
$$\left. + \frac{k_2 - k_3}{k_1 - k_2 + k_3} \exp(-(k_1 + k_3)[h\nu]t) \right) + I_0$$

### Quantitative analysis of photo-cross-linking (PXL) reaction with proteins

The model proteins were dissolved in SEC buffer (20 mM HEPES, 150 mM NaCl, pH 7.8) to a final concentration of 600 µg/mL, and were aliquoted into 100 µL. For the NHS-ester reaction with primary amine, the samples were incubated in the dark after the addition of 1 µL sulfo-SDA solution. 4 µL of 1 M Tris-HCl at pH 7.8 solution was added to the sample to quench the chemical cross-linking reaction. Photo-cross-linking reactions were performed with various optical power density $[h\nu]$ and irradiation time $t$. Immediately after irradiation, each sample was added with 1 mL pre-chilled acetone and placed at $-20\,°C$ overnight to fully precipitate the protein, which was then collected by centrifugation at 15,000 rpm for an hour.

The precipitate was dissolved in 8 M urea and 0.1 M Tris-HCl at pH 8.5, reduced with 5 mM DTT at 25 °C for 10 min and alkylated with 10 mM iodoacetamide in the dark for 15 min. Subsequently, 3 volumes of Tris-HCl were added to the sample, which also contained 1 mM $CaCl_2$ (to suppress chymotrypsin activity) and 20 mM methylamine (to reduce carbamate modification at the N-terminus of the peptide). Trypsin digestion was carried out at 37 °C overnight with sequencing-grade trypsin (Promega, diluted at a mass ratio of 1:20). The reaction was quenched with trifluoroacetic acid at a final concentration of 5%.

Trypsin-digested peptides were purified with C18 spin tips (Thermo Fisher) and were analyzed on the Orbitrap Fusion Lumos mass spectrometer (Thermo Fisher) coupled to an EASY-nLC 1200 liquid chromatography system, with a 75 µm, 2 cm Acclaim PepMapTM 100 column. The peptides were eluted using a 65 min linear gradient from 95% buffer A (water with 0.1% formic acid) to 35% buffer B (acetonitrile with 0.1% formic acid) at a flow rate of 200 nL/min. Each full MS scan (at a resolution of 70,000) was followed by 15 data-dependent MS2 scans (at a resolution of 17,000), with high-energy collisional dissociation set to 30 and an isolation window of 1.6 m/z. Precursors of charge state ≤3 were collected for MS2 scans in the enumerative mode; precursors of charge states of 3–6 were collected for MS2 scans in the cross-link discovery mode. Mono-isotopic precursor selection was enabled, and a dynamic exclusion window was set to 30 s.

The cross-linking data were analyzed with pLink2[32]. The following search parameters were used: MS1 accuracy = ± 20 ppm, MS2 accuracy = ± 20 ppm, enzyme = trypsin (with full tryptic specificity but allowing ≤ 3 missed cleavages), cross-linker = SDA (with Lys one of the cross-linked residues); fixed modifications = carbamidomethylation on

cysteine; variable modifications = oxidation on methionine and acetylation at the N-terminus. A false discovery rate of < 5% was used. The α-fragment with an additional cleavage of the cross-linked peptide at the Lys isopeptide bond was identified manually.

Both the cross-linked residues/sites and the number of peptide-spectrum match (PSM) were used to compute the relative abundance of the PXLs. PXL experiments were repeated at least three times for each protein, and only cross-links that were identified in all three experiments were used for statistics. PXL data have been deposited at the ProteomeXchange Consortium (https://www.ebi.ac.uk/pride/) via the PRIDE partner repository, identifier PXD048452.

Cartesian distances between the Cα atoms of cross-linked residues in each PSM was computed from the known structures, assuming the proteins are strictly monomeric. The Solvent accessible surface area was calculated by the Python package freesasa[76]. Structural figures were illustrated with PyMOL (version 3.6, Schrödinger LLC).

### Reporting summary
Further information on research design is available in the Nature Portfolio Reporting Summary linked to this article.

## Data availability
The cross-link data generated in this study have been deposited in the ProteomeXchange Consortium (PRIDE) database under accession code PXD048452, and the tabulated CSV file for the identified PXLs for the test proteins upon irradiation at 35 mW/cm op for 10 min also provided in the Source Data file. All other data is available from the corresponding author upon request. Source data are provided with this paper.

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

## Acknowledgements

We thank the National Center for Protein Sciences at Peking University in Beijing, China, for assistance with mass spectrometry experiments. We thank Profs. Meng-Qiu Dong and Jianbo Wang for stimulating discussions, and thank Prof. Meng-Qiu Dong for the kind gift of GFP protein. This work has been supported by grants from the National Key R&D Program of China (2023YFF1204400 to C.T.) and the National Natural Science Foundation of China (92353304 to C. T. and 22161132013 to C.T.).

## Author contributions

Y. J. and C. T. designed the project, Y. J. and H. N. set up the MRM-MS analysis instrument, and Y. J. collected the data with the assistance of NMR and MS from H. N., S. D., H. F., Xiu. Z., and L. W. Y. J., Xinghe. Z. and J. F. analyzed the data, and Y. J. and C. T. wrote the manuscript, with comments from all other authors.

## Competing interests

C.T., Y.J., and H.N. have filed a Chinese utility model patent application (ZL2024207219076) for the "Multidimensional In-line Photo-reaction Monitoring System" described in this work. All other authors declare no competing interests.
