## [Peer Review File · Nature Communications]

Dissecting Diazirine Photo-reaction Mechanism for Protein Residue-Specific Cross-linking and Distance MappingREVIEWER COMMENTS

Reviewer #1 (Remarks to the Author):

This manuscript presents an in-depth investigation of the photo-reaction mechanism of alkyl diazirines and the implications for photo-crosslinking experiments. The authors have developed a real-time photo-reaction system coupled with mass spectrometry monitoring, which allows them to tease apart the pathways involved in diazine photocrosslinking. The findings provide valuable insights into the distinct reactivity of the diazo and carbene intermediates towards different protein residues and demonstrate how tuning the reaction conditions can enhance the selectivity for polar residues. I find the results convincing and the analysis thorough.

The scope of the study is narrowly focused on a single, popular, diazine compound and the generalizability of the conclusions to other diazine-containing probes remains to be established. The authors could comment on the applicability to diazine-containing amino acids or fluoroalkyl diazirines.

Were all of the tripeptides equally soluble and therefore available to be crosslinked?

On the proteins, the authors should more thoroughly address potential issues with the assignment and validation of the observed cross-linked sites, especially for the less reactive nonpolar residues. For the distance restraints, was it assumed that all of the proteins would be crosslinked as monomers? Some form oligomers and should be annotated as such. A figure with these crosslinks displayed on the used structure/model would be useful.

It isn't indicated how many datapoints are produced to generate Figure 3b or Figure 4.

Is there any evidence of the reaction of SDA with the solvents?

Minor issues:

Figures:

The authors should use more colors in the figures. Colors overlap for the carbene, the carbonyl, or the formula, so it isn't always clear what is referenced.

Figure 3 d – the equation on the y-axis doesn't match the legend.

Figure 5 - While Figure 5b doesn't indicate how many residues are surface accessible a theoretical surface accessibility based on known structures would also be useful.

Figure 5c, what does it mean to have negative distances?

Figure 5d, I don't understand what the figure shows. Is there a better way to show solvent accessibility of residues?

Figure 5e, the annotation is unreadable.

Figure 5f, crosslinks need to be much clearer on the structure.

Text:

“As the hydrophobic residues are poorly reactive with alkyl diazirine through the carbene mechanism, especially upon water exposure (Fig. 3e and Fig. S9), a possible explanation is that these residues are incorrectly assigned using the search engine designed for CXL. Together, the PXLs involving polar residues are largely consistent with protein structures (Fig. 5c).” - A score distribution along with decoys is needed to demonstrate this point.

“side-chain of the cross-linked peptide (Fig. 5e and Fig. S14).” - This should be calling Figure S15.

Reviewer #2 (Remarks to the Author):

In this manuscript, Tang and co-workers closely examined the photolysis and photo-reaction mechanisms of alkyl-diazirine. By using a photo-reactor with a real-time MS analysis system, they identified the differential mechanisms of diazirine reaction with different amino acids, and further performed MS analysis on proteins using diazirine-based hetero-bifunctional crosslinkers. The mechanistic study section is of significant interest and is potentially very helpful for future applications of diazirine scaffolds. However, there are several points that need clarification or further demonstration:

Major edits:

- After demonstrating the photolysis mechanism, the authors went directly into photoreaction studies with different tripeptide scaffolds without explaining why tripeptide specifically was chosen here. Have the authors monitored reactions with other types of substrates such as single amino acids?
- The authors' results demonstrating the reaction mechanisms with different amino acids are very interesting and could be useful for the general audience. It would be helpful if the authors provide a summary table/figure to summarize their discovery between line 185-196 and point out the possible explanation of each outlier.
- The authors used increasing water content to monitor the reactivity of diazirine with AEA and AAA tripeptides in Fig. S9. However, since only one example for either polar or non-polar categories was shown here, it's hard to directly come to a conclusion from this data only that "carbene intermediate is more susceptible to water quenching" (line 220). It will be helpful if the authors can include more examples, if not the full panel of amino acids of interest here.
- In line 271, the authors attempted to explain why SDA photo-xlinked Gly, Ala and Ile residues at a distance exceeding 10A. They stated an possible explanation that "these residues are incorrectly assigned using the search engine designed for CXL." The same statement came up for Thr in line 276. It raises serious concern as for how confident the other datapoints collected within 10A as well as with other amino acids are. Has the authors tried different approaches to validate the identified XL sites?

Minor edits:

- In line 42, the authors explicitly mentioned that "diazirine's photo reaction is triggered

upon the irradiation at a wavelength of ~365 nm, which incurs little photochemical damages to biological samples.” Though ref 13-21 were all using diazirine for biological studies, it’s commonly believed that uv light has cytotoxicity. Blue light was even deemed not ideal for cellular experiments. It is suggested that the authors check this statement.

- When the authors demonstrate that water content can affect diazirine reactivity with amino acids, did the authors use only water and not some buffer? It would be helpful to clarify. If the authors have tested the affect of pH or different buffer, it would be also helpful to include that data

- It is interesting that the authors mentioned “distance mapping against the protein structure” in line 289 and showed the structure of BSA with crosslinks annotated in Fig 5f. I think it’s a great way to visualize the findings in this manuscript on the protein structure. However, it’s not clear what amino acids are labeled for each crosslinks. It is also hard to see clearly the distance numbers. Also, since there are 9 proteins tested, it would be good to include annotated models for the other proteins.

- For the photo-xlinking part, it would be helpful if the authors can provide the identified peptide information, including their assignments and PSM counts, in a table format.

Reviewer #1

This manuscript presents an in-depth investigation of the photo-reaction mechanism of alkyl diazirines and the implications for photo-crosslinking experiments. The authors have developed a real-time photo-reaction system coupled with mass spectrometry monitoring, which allows them to tease apart the pathways involved in diazirine photocrosslinking. The findings provide valuable insights into the distinct reactivity of the diazo and carbene intermediates towards different protein residues and demonstrate how tuning the reaction conditions can enhance the selectivity for polar residues. I find the results convincing and the analysis thorough.

The scope of the study is narrowly focused on a single, popular, diazirine compound and the generalizability of the conclusions to other diazirine-containing probes remains to be established. The authors could comment on the applicability to diazirine-containing amino acids or fluoroalkyl diazirines.

Diazirine-containing amino, most notably, photo-Leu or photo-Lys, will have the same reactivity as we discussed in the manuscript, but provide more stringent distance restraint. However, there are some trade-offs. First, diazirine-containing amino acid can be introduced metabolically but may perturb the proper protein folding, while site-specific labeling with orthogonal codon usage may have a low incorporation level. Second, Lys residues are located at or near protein surface, whereas Leu residues are more likely located in the hydrophobic core. Therefore, substitution with photo-Leu may perturb protein structure, and upon irradiation, the diazirine group may only react with neighboring hydrophobic residues through carbene mechanism, affording heterogeneous and mostly uninterpretable cross-linking data. We have expanded the related Discussion in the revised manuscript.

Among fluoroalkyl diazirines, 3-phenyl-3-(trifluoromethyl)diazirine, also known as TPD, was considered one of the most suitable photophores for photoaffinity labeling. But these types of diazirines suffer from bulkiness and hydrophobicity, and are less opted nowadays in biological applications. The fluorine substitution draws the electrons, which makes the diazo intermediate unreactive and quickly convert to carbene. Therefore, fluoroalkyl diazirines have been shown to react with proteins exclusively via the carbene mechanism (*Eur. J. Org. Chem.*, 2008: 2513-2523. <https://doi.org/10.1002/ejoc.200701069>). In the revised manuscript we have rewritten this sentence in the Introduction and incorporated additional references.

Were all of the tripeptides equally soluble and therefore available to be crosslinked?

The tripeptides were first dissolved in DMSO and were prepared in 1 μ M final concentration for cross-linking, while their solubility should be much higher. We have now emphasized this in the Methods of the revised manuscript.

On the proteins, the authors should more thoroughly address potential issues with the assignment and validation of the observed cross-linked sites, especially for the less reactive nonpolar residues.

Our study shows that unambiguous assignment can be obtained by maximizing the yield of the diazo intermediate, which preferentially reacts with polar and charged protein residues. The unique photo-adduct could be confirmed by the MS fragmentation pattern (**Fig. 5b** and **Fig. S12** in the revised manuscript) and by NMR spectroscopy (**Fig. S7**). There are two issues with the assignment of cross-linkings involving nonpolar residues—first, the absolute yield of carbene intermediate is lower than that of diazo intermediate, and second, carbene reaction gives rise to a mixture of products, even for the tripeptides. A typical NMR spectrum of the SDA adduct is shown below, shown as **Fig. S8** in the revised manuscript.

Fig. S8 SDA photo-reaction to AIA yields heterogeneous adducts, with the new NMR peaks indicated by black arrows.

Though the established software for CXL analysis can tentatively assign the cross-linked non-polar residue, other non-polar and polar residues in the peptide are also possible (if not equally possible). In the revised manuscript, we have also introduced the precursor mass error (PME), which is indicative of possible mis-assignment for PXLs involving non-polar residues. Shown in **Fig. S16** of the revised manuscript, larger and more scattered PME values were observed for cross-links involving nonpolar residues.

Fig. S16 Precursor mass errors (PME) for the cross-linked peptides involving polar (red) and nonpolar (blue) residues.

For the distance restraints, was it assumed that all of the proteins would be crosslinked as monomers? Some form oligomers and should be annotated as such. A figure with these crosslinks displayed on the used structure/model would be useful.

Yes, the majority of the cross-linking data can be accounted for with the monomer structure. For example, though some of the cross-links for AdK (test protein 9) can be considered overlength, the intermolecular cross-link distance calculated based on the dimer structure (built from crystal symmetry) are even longer. We have also checked protein oligomerization status after photo-cross-linking using SDS-PAGE. As shown below, no oligomeric bands were observed for the proteins tested.

Fig. X2 SDS-PAGE analysis of the proteins upon sulfo-SDA photo-cross-linking.

Nevertheless, for proteins that are dimeric, diazirine photo-cross-linking can readily stabilize such interaction. For example, a re-analysis of the photo-cross-linking data presented in the paper by Wahl and coworkers (*Nat Commun*, 2020, 11, 6418, <https://doi.org/10.1038/s41467-020-20159-3>) revealed that the inter-molecular cross-links involving charged/polar residues are mostly consistent with their cryoEM structure (but not for the non-polar residues).

In the revised manuscript, we have clarified this in the Methods for the analysis of protein PXLs. We have mapped the assigned cross-links to the structures of the proteins we have tested, which is shown as **Fig. S18** in the revised manuscript.

It isn't indicated how many datapoints are produced to generate Figure 3b or Figure 4.

The curves shown in **Fig. 3b** are obtained from theoretical fittings. There are five possible elementary reactions for **HY** photo-reaction with diazine (see **Supplementary Note**). Upon the production of diazo intermediate **B**, the consequent reaction with **HY** tripeptide can arise from these four mechanisms, as represented with functions of S23, S24, S25, and S26. The carbene mechanism is directly related to the production of **C**, with the corresponding equation shown as S31.

Thus, the production of **SDA-HY** can be accounted for with a linear combination of these five elementary reactions. **Fig. 4** show the theoretical 2D plots, once we have determined the relative contributions of these mechanism for each type of residue (**Table S1**). We have made further clarification in the main text and figure legends of revised manuscript. We have also added additional explanations in the **Supplementary Note**.

Is there any evidence of the reaction of SDA with the solvents?

Upon photolysis, both diazo and carbene intermediates can be quenched by water, as shown in **Fig. S10**. This property also makes diazine photo-cross-linking less likely (than BS₃) to (artificially) cross-link protein oligomers. We have also performed additional water competition data, which shows carbene intermediate can be equally if not more likely quenched. It has previously been reported that the carbene intermediate derived from aryl diazine can be quenched acetone, pyridine, and other organic solvent (Adamsu, et. al, *J. Chem. Soc., Perkin Trans.* 1998, 2, 1093-1100, <https://doi.org/10.1039/A707586C>).

Fig. S10 | The absolute yield of polar and non-polar residues decreases with increasing water content (v/v) in the solvent.

Minor issues:

Figures:

The authors should use more colors in the figures. Colors overlap for the carbene, the carbonyl, or the formula, so it isn't always clear what is referenced.

More conspicuous labeling is used for **Fig. 1** and **Fig. 3**, especially involving the chemicals.

Figure 3d – the equation on the y-axis doesn't match the legend.

Thank you for pointing this out. Also to make the plot clearer, we have changed the Y-axis label to “diazo/carbene contribution”.

Figure 5 - While Figure 5b doesn't indicate how many residues are surface accessible a theoretical surface accessibility based on known structures would also be useful.

Relative solvent-accessible surface area (RASA) of the cross-linked residues of the subject proteins is shown in **Fig. S17** of the revised manuscript.

Figure 5c, what does it mean to have negative distances?

Since diazirine reacts with a multitude of protein residues, we had to normalize the distances by subtracting the respective maximally possible distance for each type of residue (**Fig. S13** of the revised manuscript). A negative value means the calculated distance of a PXL is shorter than (within) the maximally allowed theoretical distance. We have made clarification in the revised manuscript.

Figure 5d, I don't understand what the figure shows. Is there a better way to show solvent accessibility of residues?

Fig. 5d is a carved-in representation of the cross-link involving K245 and K297, with these two residues shown as balls-and-sticks and the rest of the protein shown as cartoon-and-surface. We have enlarged the structural figure and added explanation in the legend.

Figure 5e, the annotation is unreadable. Figure 5f, crosslinks need to be much clearer on the structure.

In the revised manuscript, we have removed labels and make PXLs clearer.

Text:

“As the hydrophobic residues are poorly reactive with alkyl diazirine through the carbene mechanism, especially upon water exposure (Fig. 3e and Fig. S9), a possible explanation is that these residues are incorrectly assigned using the search engine designed for CXL. Together, the PXLs involving polar residues are largely consistent with protein structures (Fig. 5c).” - A score distribution along with decoys is needed to demonstrate this point.

Even on the peptide level, the diazirine reaction to the nonpolar residues yielded a mixture of products that cannot be easily analyzed. For protein cross-linking data, we have employed matched peptide spectra (**Fig. S11**), relative solvent accessible area (**Fig. S17**), and MS2 fragmentation (**Fig. 5b** and **Fig. S12**) to ascertain the assignment of the PXLs involving

polar/charged residues, which affords cross-linking distances largely consistent with the protein structures (**Fig. S14** and **Fig. S15**). In addition, **Fig. S16** in the revised manuscript, shows the evaluation of PME values, illustrating larger errors associated with the assignment of nonpolar cross-links.

The potential mis-assignment of the cross-links involving nonpolar residues can be construed this way. Carbene intermediate has equal likelihood to be inserted into C-H (or N-H) bonds of nonpolar residues, as well as polar residues. Thus, though we can “assign” diazirine cross-link to a particular nonpolar residue, we cannot rule out the possibility, chemically, the cross-linking/insertion to a neighboring residue, be it nonpolar or polar. Moreover, the exact positions of carbene insertion into the “assigned” nonpolar residue is also heterogeneous.

The instrumental set-up described in our current manuscript allowed us to maximize the yield of diazo intermediate. Thus, the diazo intermediate is enriched for the reactions with polar/charged residues, even though there can be some contribution from the carbene intermediate. We have incorporated additional discussion in the revised manuscript.

“side-chain of the cross-linked peptide (Fig. 5e and Fig. S14).” - This should be calling Figure S15.

Thank you for pointing out. We have incorporated additional Supplementary figures and made sure the numberings are ordered.

Reviewer #2 (Remarks to the Author):

In this manuscript, Tang and co-workers closely examined the photolysis and photo-reaction mechanisms of alkyl-diazirine. By using a photo-reactor with a real-time MS analysis system, they identified the differential mechanisms of diazirine reaction with different amino acids, and further performed MS analysis on proteins using diazirine-based hetero-bifunctional crosslinkers. The mechanistic study section is of significant interest and is potentially very helpful for future applications of diazirine scaffolds. However, there are several points that need clarification or further demonstration:

Major edits:

After demonstrating the photolysis mechanism, the authors went directly into photoreaction studies with different tripeptide scaffolds without explaining why tripeptide specifically was chosen here. Have the authors monitored reactions with other types of substrates such as single amino acids?

We used small peptides instead of free amino acids for these reasons. Appending Ala residues on the N- and C-terminal ends of the residue mimics a real protein. Moreover, we have modified the N- and C-terminal Ala residues of the resulting peptides, which blocks the reactive polar groups. The peptides are commercially available, whereas the protected amino residues alone are very expensive. Ala-padded peptides have been used for the evaluations of backbone dihedral angles (*PNAS*, 2004 101, 27, 10054-10059, <http://doi.org/10.1073/pnas.2301064120>; *PNAS* 2008, 105, 34, 12259-12264; <https://doi.org/10.1073/pnas.0706527105>), which are now cited in the revised manuscript. On the other hand, as we show in our current work, Ala has a quite low reactivity with diazirine (in comparison to polar/charged residues), which therefore can be considered as a background. We have added explanation in the revised manuscript.

The authors' results demonstrating the reaction mechanisms with different amino acids are very interesting and could be useful for the general audience. It would be helpful if the authors provide a summary table/figure to summarize their discovery between line 185-196 and point out the possible explanation of each outlier.

The relative contributions of the elementary mechanism for the 20 amino acids (in tripeptides) are shown in **Table S1** (with standard deviations added). There are five possible elementary reactions for **HY** photo-reaction with diazirine (see **Supplementary Note**). Upon

the generation of diazo intermediate **B**, the consequent reaction with **HY** tripeptide can arise from these four mechanisms, as represented with functions of S23, S24, S25, and S26, with the weighting factors of b_1 , b_2 , b_3 , and b_4 . The carbene mechanism is directly related to the production of **C**, with the corresponding equation shown as S31, with the weighting factor of c . The fitted values of b_1 , b_2 , b_3 , b_4 , and c are indeed very interesting. For example, a significant proportion of Pro reacts with diazirine via the diazo mechanisms, which can be attributed to the acidity of α -carbon. On the other hand, Gln and Asn react with SDA primarily through the carbene mechanism, which is completely different from Glu and Asp residues.

The authors used increasing water content to monitor the reactivity of diazirine with AEA and AAA tripeptides in Fig. S9. However, since only one example for either polar or non-polar categories was shown here, it's hard to directly come to a conclusion from this data only that "carbene intermediate is more susceptible to water quenching" (line 220). It will be helpful if the authors can include more examples, if not the full panel of amino acids of interest here.

We have performed additional tripeptide photo-reactions in the presence of water, as shown in **Fig. S10** of the revised manuscript. You are right — as the production of carbene intermediate is already low in comparison to the diazo intermediate (using our irradiation setup), and it is hard to say which is more susceptible to water quenching. We have rewritten the sentence, and stated that "the photo-adduct with polar residues generally gives higher yield than that with non-polar residues".

Fig. S10 | The absolute yield decreases with increasing water content (v/v) in the solvent.

In line 271, the authors attempted to explain why SDA photo-xlinked Gly, Ala and Ile residues at a distance exceeding 10Å. They stated an possible explanation that “these residues are incorrectly assigned using the search engine designed for CXL.” The same statement came up for Thr in line 276. It raises serious concern as for how confident the other datapoints collected within 10Å as well as with other amino acids are. Has the authors tried different approaches to validate the identified XL sites?

The possible misassignment of cross-links involving nonpolar residues can also be construed this way. Carbene intermediate has equal likelihood to be inserted in C-H (or N-H) bonds of nonpolar residues, as well as polar residues. Though we can tentatively assign diazirine cross-link to a particular nonpolar residue, we cannot rule out the possibility, chemically, the cross-linking/insertion to a neighboring residue, be it nonpolar or polar. Moreover, the exact positions of carbene insertion into the “assigned” nonpolar residue is heterogeneous.

The reactivity of SDA towards Thr is not particularly high in DMSO (**Fig. 3e**), which is

likely due to steric hindrance from the vicinal methyl group. The reaction between the diazo intermediate and hydroxyl group is highly susceptible to water quenching (**Fig. S10**), while the Thr PXLs identified in the test proteins are highly solvent-exposed (**Fig. S17**). As we aim to illustrate how to take the cream of the crop while leaving out the potentially false positive information of diazirine cross-links, in the revised manuscript we suggest to use PXLs of Glu, Asp, Tyr, and Ser for structural restraints.

Fig. S16 Precursor mass errors (PME) for cross-linked peptides involving polar/nonpolar residues.

For protein cross-linking data, we have employed matched peptide spectra (**Fig. S11**), relative solvent accessible area (**Fig. S17**), and MS2 fragmentation (**Fig. 5b** and **Fig. S12**) to ascertain the assignment of the cross-links involving polar/charged residues. In addition, we have incorporated **Fig. S16** in the revised manuscript, which evaluates the PME values and illustrates larger errors associated with nonpolar cross-links. For the test proteins we used (with known structures), the $C\alpha$ - $C\alpha$ distance compatibility (**Fig. S11-S13**) constitutes one last test for the correct assignment of the photo-cross-links.

In line 42, the authors explicitly mentioned that “diazirine’s photo reaction is triggered upon the irradiation at a wavelength of ~365 nm, which incurs little photochemical damages to biological samples.” Though ref 13-21 were all using diazirine for biological studies, it’s commonly believed that uv light has cytotoxicity. Blue light was even deemed not ideal for cellular experiments. It is suggested that the authors check this statement.

365-nm wavelength belongs in the UVA range and is much less harmful than UVB and UVC lights. In fact, 365-nm light has also been used for skin tanning. There are much fewer (except for diazirine) functional groups in a protein adsorbing 365-nm light (triggering subsequent photo-reactions) than UVB light. For *in-situ* cross-linking of cells, proteins and protein complexes are structurally fixed upon photo-cross-linking but with the chemical entity of the molecules intact, which is immediately followed by cell lysis.

When the authors demonstrate that water content can affect diazirine reactivity with amino acids, did the authors use only water and not some buffer? It would be helpful to clarify. If the authors have tested the affect of pH or different buffer, it would be also helpful to include that data.

Thank you for pointing this out — we are carrying out additional studies to further increase the utilization of diazo reaction with the variations of wavelength and buffer pH. While the tripeptides react with diazirine in DMSO or water/DMSO mixture, the proteins were prepared in the HEPES buffer, as described in the Methods. The contribution of the five elementary reactions, in particular proton-catalyzed b_3 and b_4 mechanisms, can be affected by the pH. We have added this in the discussing the mechanism in the Supplementary Note of the revised manuscript.

It is interesting that the authors mentioned “distance mapping against the protein structure” in line 289 and showed the structure of BSA with crosslinks annotated in Fig 5f. I think it’s a great way to visualize the findings in this manuscript on the protein structure. However, it’s not clear what amino acids are labeled for each crosslinks. It is also hard to see clearly the distance numbers. Also, since there are 9 proteins tested, it would be good to include annotated models for the other proteins.

We have made **Fig. 5f** clearer (with the distance label removed). We have also mapped the identified cross-links (involving polar/charged residues) for all the proteins we have tested, shown as **Fig. 18** of the revised manuscript. We did not label the distances (and a sperate Table is now uploaded), which would be hard to see.

For the photo-xlinking part, it would be helpful if the authors can provide the identified peptide information, including their assignments and PSM counts, in a table format.

The PXL data have been deposited at the ProteomeXchange Consortium (<https://www.ebi.ac.uk/pride/>) via the PRIDE partner repository, with the accession code of PXD048452. We have also uploaded a Table as additional SI, which contains the detailed PXL information.

REVIEWERS' COMMENTS

Reviewer #1 (Remarks to the Author):

The authors' responses and changes have made the manuscript clearer and more useful to the field.

Reviewer #2 (Remarks to the Author):

In the edited manuscript, the authors made an effort providing extra information and clarifying their statements. In particular, the new Fig. S10 demonstrating how water content affect reactivity is very interesting. It is very clear how differently the reactions went for different amino acids from this direct comparison. A minor edit is to include the conversion rate number on the bar graphs since the absolute number for AAA, AQA and AIA is low and hard to see. Another point of consideration is the statement of 365 nm uv light incurring “little photochemical damages to biological samples” as one of the first sentences in the introduction section. Since the authors use this statement as the reason for biochemical studies (see line 43, “as a result, diazirine chemistry has been employed in a multitude of biological applications...”, it is a bit misleading or at least exaggerating to emphasize the bio-safety of uv light here. This is more true so given that the definition of “biological samples” may vary given the wide audience of this journal. It has little to do with what the authors replied, such as UVB or UVC (they are not mentioned in the text at all), “skin tanning” (from the authors’ reply), or how the samples were prepared. I suggest taking this comment out from the introduction section to avoid confusion.

Reviewer #2 (Remarks to the Author):

In the edited manuscript, the authors made an effort providing extra information and clarifying their statements. In particular, the new Fig. S10 demonstrating how water content affect reactivity is very interesting. It is very clear how differently the reactions went for different amino acids from this direct comparison. A minor edit is to include the conversion rate number on the bar graphs since the absolute number for AAA, AQA and AIA is low and hard to see. Another point of consideration is the statement of 365 nm uv light incurring “little photochemical damages to biological samples” as one of the first sentences in the introduction section. Since the authors use this statement as the reason for biochemical studies (see line 43, “as a result, diazirine chemistry has been employed in a multitude of biological applications...”, it is a bit misleading or at least exaggerating to emphasize the bio-safety of uv light here. This is more true so given that the definition of “biological samples” may vary given the wide audience of this journal. It has little to do with what the authors replied, such as UVB or UVC (they are not mentioned in the text at all), “skin tanning” (from the authors’ reply), or how the samples were prepared. I suggest taking this comment out from the introduction section to avoid confusion.

Response:

We have changed the fonts in Figure S10 to make them readily visible.

We have removed the statement “*little photochemical damages to biological samples*” from the sentence in the opening paragraph of Introduction.